# Evaluating the Impact of Medication Risk Mitigation Services in Medically Complex Older Adults

**DOI:** 10.3390/healthcare10030551

**Published:** 2022-03-16

**Authors:** Hubert Jin, Sue Yang, David Bankes, Stephanie Finnel, Jacques Turgeon, Alan Stein

**Affiliations:** 1Office of Healthcare Analytics, Tabula Rasa HealthCare, Moorestown, NJ 08057, USA; hjin@trhc.com (H.J.); syang@trhc.com (S.Y.); sfinnel@trhc.com (S.F.); 2Office of Translational Research and Residency Programs, Tabula Rasa HealthCare, Moorestown, NJ 08057, USA; dbankes@trhc.com; 3Precision Pharmacotherapy Research and Development Institute, 13485 Veteran’s Way, Suite 410, Lake Nona, Orlando, FL 32827, USA; jturgeon@trhc.com

**Keywords:** Program of All-Inclusive Care for the Elderly, adverse drug events, medication-related problems, drug-related problems, pharmacists, medication safety, Medicare, Medicaid

## Abstract

Adverse drug events (ADEs) represent an expensive societal burden that disproportionally affects older adults. Therefore, value-based organizations that provide care to older adults—such as the Program of All-Inclusive Care for the Elderly (PACE)—should be highly motivated to identify actual or potential ADEs to mitigate risks and avoid downstream costs. We sought to determine whether PACE participants receiving medication risk mitigation (MRM) services exhibit improvements in total healthcare costs and other outcomes compared to participants not receiving structured MRM. Data from 2545 PACE participants from 19 centers were obtained for the years 2018 and 2019. We compared the year-over-year changes in outcomes between patients not receiving (control) or receiving structured MRM services. Data were adjusted based on participant multimorbidity and geographic location. Our analyses demonstrate that costs in the MRM cohort exhibited a significantly smaller year-to-year increase compared to the control (MRM: USD 4386/participant/year [95% CI, USD 3040–5732] vs. no MRM: USD 9410/participant/year [95% CI, USD 7737–11,084]). Therefore, receipt of structured MRM services reduced total healthcare costs (*p* < 0.001) by USD 5024 per participant from 2018 to 2019. The large majority (75.8%) of the reduction involved facility-related expenditures (e.g., hospital admission, emergency department visits, skilled nursing). In sum, our findings suggest that structured MRM services can curb growing year-over-year healthcare costs for PACE participants.

## 1. Introduction

The Program of All-Inclusive Care for the Elderly (PACE) provides comprehensive, supportive services to individuals older than 55 who are certified by their state to require a “nursing home level of care” [1]. The average PACE participant is about 77 years old, has six chronic comorbidities, takes several prescription medications per month, and needs help with at least one activity of daily living [2]. A central objective of PACE is to avoid long-term institutionalization by supporting independent community living [1]. To meet these goals, PACE organizations receive capitated payments (i.e., fixed-rate dollar amount per participant) from the federal government and their state’s Medicaid program [1]. Capitation permits clinicians to provide any service needed to achieve positive outcomes in the mid to long term. Conversely, capitation implies that clinicians must strive to avoid expensive, preventable problems (e.g., preventable emergency department (ED) or hospital utilization) and unnecessary services that are unlikely to achieve participant goals of care.

While meeting this goal requires a multidimensional approach [3], medication-related morbidity is a costly societal burden that is relevant to PACE. The value-based, capitated payment model means that PACE organizations are 100% at-risk for negative outcomes; thus, negative sequelae resulting from poor medication-related outcomes would have a direct negative impact on the economics of PACE organizations. For instance, a 2018 cost-of-illness model suggested that it costs about USD 2500, on average, to treat an individual who experiences treatment failure or a new medical problem after initial prescription use [4]. Medically-complex older adults—such as PACE participants—are at particularly high risk of negative economic and clinical outcomes associated with drug-related harm [1,5,6]. For instance, one study found that for every dollar spent on medications, USD 1.33 is spent to treat associated medication-related problems (MRPs) in nursing home patients [7]. Another meta-analysis found that the odds of being hospitalized for an adverse-drug event (ADE) are four times greater in older adults compared to their younger counterparts [8]. Moreover, estimates suggest that 10–30% of hospitalizations are caused by ADEs in older adults [5].

Fortunately, the literature suggests that drug-related harm can often be avoided [9]. In particular, medication risk mitigation (MRM) services could enable PACE organizations to avoid some of the costly, negative outcomes associated with medication-related morbidity. MRM encompasses a suite of clinical pharmacy services and technological solutions that aim to optimize medication use in vulnerable older adults. Specifically, MRM services and solutions include ADE risk stratification [10]; clinical decision support software (CDSS) that aids pharmacists in the optimization of medication regimens [11]; pharmacogenomic (PGx) assessments [12]; provision of expert drug information to PACE prescribers [13]; and comprehensive medication adherence support. Until now, no controlled study has evaluated the impact of MRM services on economic outcomes in PACE. Therefore, the objective of this study is to evaluate whether PACE participants receiving MRM solutions exhibit improvements in healthcare costs and other pertinent healthcare outcomes compared to similar participants who do not receive MRM.

## 2. Materials and Methods

### 2.1. Study Design, Data Source, and Approvals

This was a retrospective, naturalistic, quasi-experimental study of 2018 and 2019 administrative medical claims data. This study was granted a waiver of informed consent from an independent institutional review board.

### 2.2. Intervention Description: TRHC’s MRM Services

For prescription needs, many PACE programs partner with one pharmacy. CareKinesis, a Tabula Rasa HealthCare (TRHC) subsidiary, is a national PACE pharmacy that provides a suite of MRM solutions to 15 K participants from more than 60 PACE organizations across the US. CareKinesis provides these MRM services to complement the physical provision of prescribed medications. A detailed summary of the specific MRM solutions is provided in Table 1.

When MRM solutions are deployed into PACE pharmacy practice, pharmacists are enabled to identify MRPs and to issue recommendations to resolve them [11]. Pharmacist-provided “Medication Safety Reviews” (MSRs) are the conduit through which such interventions are delivered to PACE providers. As a formal definition, “MSRs apply the principals of pharmacodynamics, pharmacokinetics, pharmacogenomics, and chronopharmacology to enhance medication safety and prevent ADEs. Additionally, MSRs address simultaneous multidrug interactions in the context of the entire drug regimen” using the aforementioned CDSS [11].

In PACE, MSRs can vary substantially in their timing, intensity, and delivery. Regarding timing, MSRs can be retrospective or prospective. A retrospective MSR involves clinical interventions that aim to resolve MRPs for medications that have already been prescribed, dispensed, or ingested. A prospective MSR issues interventions that aim to prevent MRPs for medications that have not been ingested or dispensed yet. Regarding intensity, MSRs can aim to resolve one or more MRPs per patient. Regarding delivery, MSRs can be delivered telephonically or electronically. Telephonic delivery might involve an ad hoc call with the prescriber or a formal conference call with the PACE team to review multiple MRPs for multiple patients (operationally defined as a “polypharmacy call”). Electronic delivery could involve instant messaging through the prescription management system (EireneRx^®^, CareKinesis, Inc. and TabulaRasa HealthCare, Inc., Moorestown, NJ, USA), encrypted e-mails, or formal faxed reports.

Figure 1 summarizes the MRM solutions in a workflow diagram:

### 2.3. Subjects and Outcomes

In addition to the PACE pharmacy subsidiary (CareKinesis), TRHC also has a subsidiary that acts as a third-party administrator for several PACE organizations (CareVention HealthCare™ Third Party Administration). Therefore, TRHC has full administrative medical claims data for two types of PACE programs: (1) PACE clients that receive pharmacy (i.e., MRM services) through CareKinesis (intervention group) and (2) PACE clients that choose to receive pharmacy services elsewhere (control group). Thus, our study sample was non-randomized; all members for whom we had administrative claims were initially eligible for inclusion. Moreover, cohort selection was naturalistic since PACE organizations self-selected CareKinesis (i.e., MRM) services.

Because our intention was to compare the 2018-to-2019 changes in outcomes for both cohorts, we first excluded any participant that was not continuously enrolled during 2018 and 2019. We identified 19 PACE organizations for which (a) administrative medical records were available for the entirety of calendar years 2018 and 2019 and (b) had data use agreements that permitted retrospective research. Of these 19 organizations, 12 organizations received pharmacy services through CareKinesis (MRM), and 7 did not (no MRM).

We compared the following outcomes between the two groups:Medical costs. We evaluated the total combined facility (e.g., hospital) and physician (e.g., outpatient services, office visit) expenditures as well as each type of expenditure individually (i.e., hospital, physician). Costs were defined as the total amount that was adjudicated each year (i.e., 2018 and 2019) in US dollars in the claims data. Facility and physician costs were defined from the claim details field in the data. Facility and physician charges were encoded as UB92 and HCFA, respectively. Thus, the total costs were the sum of UB92 and HCFA.Fraction of participants with ≥1 reported ADE. ADEs were defined as any A- or B-level International Classification of Diseases, Tenth Revision, Clinical Modification (ICD-10) code as defined previously by Hohl et al. [29].Fraction of participants with ≥1 fall. Falls were defined using the following W-group ICD-10 codes: 01, 03–11; 17–19 as well as R29.6.Number of ED visits and hospital admissions. Both were identified by line-item claims.

### 2.4. Analysis

PACE organizations are free to select their pharmacy provider. Organizations either select CareKinesis’ MRM services or obtain pharmacy services elsewhere. Thus, differences across relevant confounding variables could bias results. Typically, propensity score matching is deployed in observational studies as a way to adjust for potentially influential covariates [30]. However, this approach appeared less relevant since our control group was smaller than our intervention group, which limited our capability to do appropriate matching of individuals. To ensure a fair comparison between our two groups, we decided to weight our analyses using participants’ baseline hierarchical condition category (HCC) scores.

For context, HCC scores are—broadly—a marker of multimorbidity and patient acuity. The Centers for Medicare and Medicaid Services (CMS) use HCC scores to adjust annual capitation payments for individual PACE participants [31]. A summary HCC score is derived using (a) a participant’s ICD-10 codes from the previous calendar year and (b) participant demographic data (age, sex, Medicaid status, disability status) [32]. We used each participant’s December 2019 HCC score because this reflected all 2018 diagnoses. Some health services researchers have found that HCC scores can serve as a valid predictive tool for hospitalizations, ED visits, and costs in various cohorts [33,34].

We performed the risk adjustment through the following four steps:Set HCC bins with a set of boundaries.Calculate weight for each participant. Let x_i_ represent the number of participants in the i-th bin for the MRM cohort. Let y_i_ represent the number of participants in the same bin for the control cohort. Therefore, x_i_/y_i_ represents the weight to apply to all participants in the i-th bin to make the control cohort equivalent to the MRM cohort.For each bin, add a padding parameter—0.001—to avoid bins with zero participants and provide a smoothing effect.Add a normalization step to ensure that the sum of the control cohort weights equals the control cohort sample size.

The normalized weights were then used to adjust each clinical outcome for control participants at each bin. For cost-related outcomes, we made one modification to this adjustment. Geographic differences between groups could bias results because medical costs in the US can substantially differ regionally [35]. In PACE, CMS accounts for this variation by applying a county-level adjustment to the HCC score [36]. Therefore, we adjusted the cost-specific outcomes using the actual capitated rate paid by CMS for each participant.

This adjustment procedure carries two implications for exclusion criteria. First, we excluded anyone without a baseline HCC score. Second, patients with end-stage renal disease (ESRD) were excluded since they are scored using a completely different HCC model [17], making the adjustments described above impossible. Regardless of HCC, excluding ESRD is reasonable since such patients tend to consume a disproportionate amount of financial resources [37].

After making all exclusions and adjustments, we first calculated the 2018-to-2019 changes in outcomes (i.e., financial and clinical outcomes) for both cohorts. For continuous outcomes, we compared the two cohorts’ year-over-year changes (i.e., weighted mean difference) using a 2-sample *t*-test, weighted using each participant’s baseline CMS HCC score (for non-cost outcomes) or capitated payment amount (for cost outcomes). Since costs tend to have skewed distributions, we also performed the comparisons using the Wilcoxon test. For the categorical outcomes (i.e., participants with ≥1 ADE or fall), we used a chi-square test (weighted by HCC) for comparisons. This test was applied to a 3 × 2 contingency table such as that shown in Table 2. Table 2 applies to ADEs; we used a comparable table for falls.

We considered *p* values < 0.05 statistically significant. All standard errors, confidence intervals, and *p*-values were computed using techniques suitable for weighted data (e.g., Kish’s effective sample size) [38]. All analyses were conducted in R version 3.5.

## 3. Results

### 3.1. Cohort Description

The entire study consisted of 2545 PACE participants across the 19 PACE organizations. The sample was predominantly female (67.2%) with an average age of 77.0 (95% CI: 76.6, 77.3) years. The MRM and control cohorts were well-balanced across age and sex. However, patients in the MRM group had a greater level of multimorbidity, as defined by HCC (mean HCC 2.68 vs. 2.58, *p* = 0.042). The two groups also differed according to geographic distribution (*p* < 0.001). Specifically, PACE participants in the MRM group were predominantly from the Western (48.6%) and Southern (28.9%) regions of the US, whereas participants in the control cohort were largely from the Northeast (70.8%). Full participant demographics can be viewed in Table 3.

### 3.2. Outcomes

As shown in Table 4, the mean total medical costs (i.e., combined facility and physician) increased from 2018 to 2019 in both cohorts, but the increase was smaller for the MRM cohort. Specifically, the MRM group’s costs increased by a mean of USD 4386 (95% CI, USD 3040–5732) per participant, whereas the control group’s costs increased by USD 9410 (95% CI, USD 7737–11,084) per participant. This USD 5024 difference between each group’s year-over-year change was significant (*p*  < 0.001); therefore, PACE organizations using MRM consumed USD 5024 less per participant from 2018 to 2019 relative to control. As depicted in Figure 2, 75.7% (USD 3807/USD 5024) of this reduction was related to facility expenditures. As shown in Table 4, both facility and physician expenditures increased less in the MRM cohort (*p* < 0.001).

For the 2018-to-2019 changes in every other outcome (Table 5), the results directionally favored the MRM group; however, the difference between the groups was not statistically significant for any of the outcomes.

## 4. Discussion

Healthcare economists have demonstrated that US healthcare expenditures are expected to rise through 2025 [39]. Reasons are multifactorial, but the upward cost trajectory is largely attributable to disease progression in an aging population [39]. Therefore, it is not surprising that we observed a year-over-year increase in total medical costs for more than 2500 medically complex older adults enrolled in 19 PACE programs dispersed across the US. However, PACE participants who received MRM services exhibited a significantly lower year-over-year increase in costs compared to participants who were not exposed to the same services, even after adjusting for the baseline capitated rate, which accounts for differences in multimorbidity and geographical location. Relative to control, the MRM group consumed USD 5024 less per participant year-over-year, where about 75% of the reduction came from facility-related expenditures. Though we did not demonstrate statistically significant differences in year-to-year changes for other outcomes, every result was directionally in favor of MRM and was clinically important, helping to explain the cost reduction. In sum, our findings suggest that MRM services can curb growing healthcare costs in PACE.

A recent study of MRM services in Medicare beneficiaries supports this idea. In an Enhanced Medication Therapy Management (EMTM) program [40], Stein et al. examined the impact of retrospective MSRs across nearly 11,500 Medicare beneficiaries [41]. The authors found that those who received MSRs consumed USD 958 less in Medicare costs (facility plus physician) year-over-year than those who did not. Similar to the study at hand, Stein et al. also found that the overwhelming majority (90%) of the savings were due to expenses incurred at facilities. The reproducibility of the financial outcomes in a more complex PACE cohort strongly suggests that the benefits of MRM services are consequential in different patient populations.

Reproducibility is logical considering how MSRs are deployed and executed in clinical practice. First, a novel risk assessment tool, the MedWise Risk Score (MRS), facilitates intervention deployment to those who are most at risk for ADEs [42]. High-MRS PACE participants have been shown to consume more medical resources (e.g., medical costs, hospitalizations, and ED visits) and suffer from more ADEs [10]. Since scores are derived from modifiable pharmacologic risk factors found within a drug regimen, pharmacists can act to mitigate risk factors contributing to negative outcomes. Though future studies need to evaluate whether PACE pharmacists’ interventions can positively alter MRS-defined risk, the aforementioned evaluation of MSRs in EMTM indicates that this is possible as long as recommendations are implemented by prescribers [42].

Regarding clinical execution, Bankes et al. found that during MSRs, PACE pharmacists identify about two medication-related problems (MRPs) per participant, where four medication-safety related MRPs—drug interactions, adverse drug reactions, high doses, and unindicated medications—account for about 80% of all MRPs logged [11]. This distribution is highly comparable to what was seen when MSRs were deployed in the EMTM setting [42]. Others have found these types of medication-safety MRPs to be quite costly. For instance, a cost avoidance model suggested that resolving the aforementioned MRPs can avoid between USD 90–675 per occurrence [43]. Importantly, PACE prescribers accept about 80% of pharmacists’ recommendations, which suggests that MRPs are indeed being resolved; thus, MRS-defined risk is being attenuated [11].

Unlike EMTM (where MSRs are performed retrospectively, after drugs have been prescribed and ingested), MRM in PACE offers pharmacists the ability to resolve such problems prospectively, at the point of prescribing. While this is likely the biggest contributor to the reduction in costs here than what was reported by Stein et al., other ancillary components of MRM that are unavailable in the EMTM setting could also help explain this difference. For example, adherence support services, pharmacogenomic consultations, and prescriber-initiated engagement of PACE pharmacists for drug information are all expected to optimize regimens and improve outcomes [13,44,45]. Future research must evaluate which MRM components are most impactful to economic and clinical outcomes. Moreover, prospectively designed research should determine whether our observed relative cost reduction represents cost savings, cost avoidance, or a blend of both.

It is expected that healthcare costs will continue to rise [39] amid Medicaid funding restrictions [46]. Our results suggest that comprehensive MRM provides a way for PACE to address this concern. This is important because PACE appears to be inattentive to medication-related morbidity from a regulatory standpoint. Specifically, current regulations do not require pharmacists to be part of the PACE interdisciplinary team [1]. This means that a PACE center may not have on-the-ground expertise in pharmacotherapeutics, pharmacology, and/or pharmacokinetics. For such centers, MRM appears to be valuable.

The primary limitation of this analysis is self-selection bias. We attempted to make the fairest comparison possible by adjusting for participant multimorbidity and geographic location. Yet, there could be some unmeasurable or unavailable variables that confounded the results. For instance, we were not able to consider the length of PACE enrollment. Next, generalizability may be limited for two reasons related to potential sampling bias. First, we only had access to administrative claims of 2500 participants from 19 PACE organizations. As of June 2021, this represents <5% of the entire PACE census and <14% of all programs [47]. Nevertheless, we had representation from various locations throughout the US, and our sample appeared similar across some demographics (e.g., age, sex) reported by the National PACE Association [2]. Second, with only seven centers in the control group, it is possible that we did not capture PACE organizations that have robust clinical pharmacy services. Still, those services would not be using the same clinical decision support systems. It is possible that the effect of MRM could be attenuated when compared against programs with robust clinical pharmacy service offerings [46,48,49,50]. Finally, we were unable to demonstrate statistically significant differences between groups for their year-over-year change in the proportion of patients with ADEs. Therefore, it is impossible to conclusively tie cost reduction to improvements in medication safety; medication safety is the main purpose of MRM. Our lack of significance is likely explained by our reliance on ICD-10 codes. Specifically, ADEs tend to be grossly underreported in administrative claims [51]. Therefore, our sample size was likely insufficient to detect significant differences for this outcome. Future studies should use a formal sample size calculation (which will necessitate a larger sample size) to draw more reliable conclusions about this important outcome.

## 5. Conclusions

In sum, PACE participants who received MRM services exhibited a smaller year-over-year increase in costs compared against risk-adjusted participants who were not exposed to MRM. Specifically, those who received MRM consumed USD 5024 less in total medical costs year-over-year than those who did not. Therefore, MRM appears to be effective at curbing rising healthcare costs in PACE.

## Figures and Tables

**Figure 1 healthcare-10-00551-f001:**
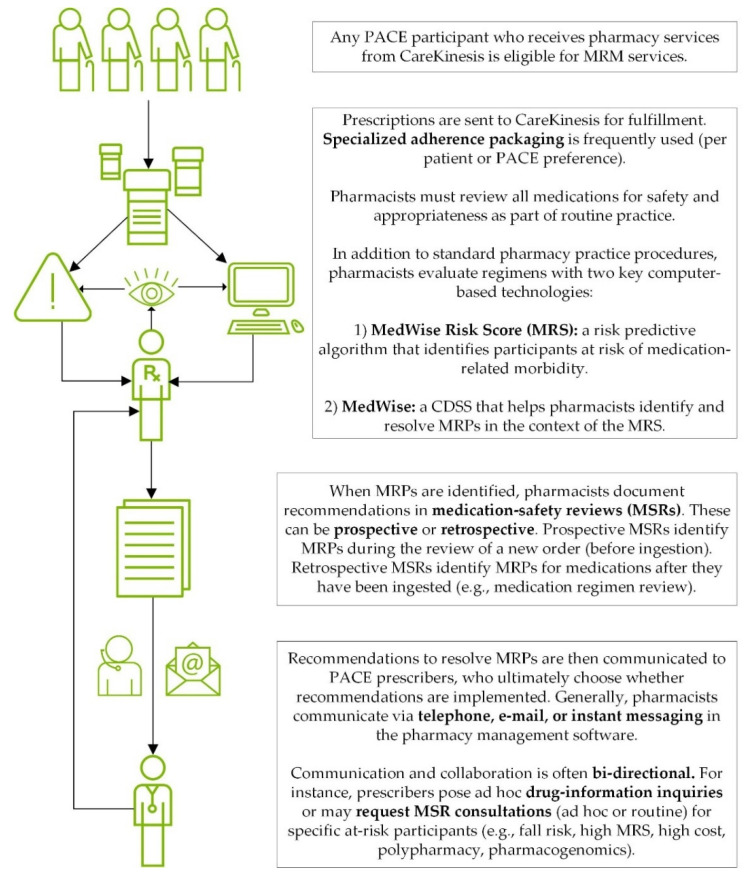
MRM workflow. Abbreviations: CDSS = clinical decision support software; MRM = medication risk mitigation, MRP = medication-related problem; PACE = The Program of All-Inclusive Care for the Elderly.

**Figure 2 healthcare-10-00551-f002:**
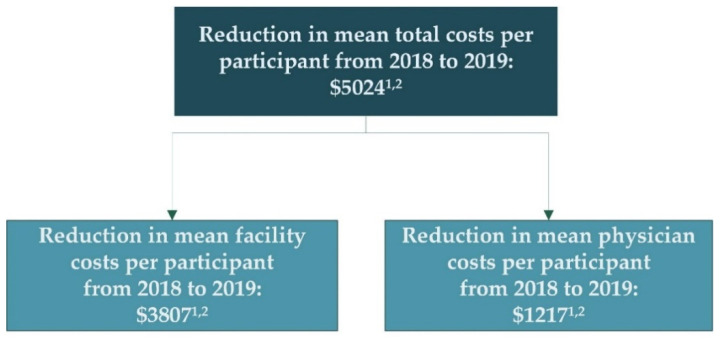
Breakdown in year-over-year cost reduction seen in MRM relative to control. ^1^ Cost reduction represents the difference between the intervention and control group’s 2018-to-2019 change in medical costs. Adjustments were applied to the control group’s 2018 and 2019 costs. Adjustments were based on the actual capitated rate. This ensured that geographic differences between MRM and control did not bias outcomes. ^2^ Denotes a statistically significant (*p* < 0.05) difference between MRM and control groups.

**Table 1 healthcare-10-00551-t001:** Summary of TRHC’s key medication risk mitigation (MRM) components in PACE.

MRM Component	Detailed Description
MedWise risk score (MRS)	Risk assessment tool that helps identify PACE participants at high risk of ADEs and in need of risk-mitigating interventions. Constructed from 5 modifiable risk factors derived from a drug regimen’s PK and PD characteristics [14,15]. Scored from 0 to 53; ≥20 is considered high risk [14]. Among PACE participants served by CareKinesis, mean MRS is 18.5 [10]. In PACE, each point rise in MRS is associated with: 8.6% increase in odds of ADEs; USD 1037 in annual medical spending; 3.2 and 2.1 additional ED visits and hospitalizations, respectively, per 100 participants per year [10]. Results confirmed in other settings, demonstrating additional associations with mortality and falls [16,17].
MedWise	This is an advanced CDSS used by TRHC pharmacists to assist clinical interventions [18]. It presents visualizations of a medication regimen within context of MRS risk factors and allows for identification of simultaneous multidrug interactions [11]. By working in tandem with the MRS, pharmacists can identify MRPs that contribute to ADEs. Visuals of Medwise abound in the literature [19,20,21,22,23].
Pharmacogenomics services (PGx)	CareKinesis PACE Pharmacy offers PGx testing with clinical interpretation/intervention for PACE programs choosing to further personalize their participants’ medication regimens [12]. CDSS ingests PGx results to help pharmacists identify drug-induced phenoconversion [24]. Thus, pharmacists can interpret consensus guidelines (e.g., CPIC) in the context of the entire drug regimen [25]. PGx services identify 2.5–3.0 gene-based interactions per PACE participant [26].
Drug information support	Clinical pharmacists provide expert advice to prescribers needing drug information prior to making a clinical decision [13]. PACE prescribers ask TRHC pharmacists a heterogeneous array of questions related to medication management. Prescribers implement about 80% of answers within drug regimens [13].
Comprehensive adherence support	TRHC’s dispensary can provide participant medications in customized adherence packaging. Refills for chronic medications are synchronized and dispensed automatically on a regular cycle basis.
Staff competency	As a condition of employment: ○ Pharmacists must be (or become) board-certified in geriatric pharmacotherapy (i.e., BCGP) [27] and certified to use the proprietary CDSS (i.e., Certified MedWise Advisor^TM^ pharmacists). Pharmacy technicians must have (or obtain) the Certified Pharmacy Technician Credential (i.e., CPhT) [28].
Medication safety review (MSR)	A service performed by pharmacists. By applying MRM components, pharmacists identify MRPs and provide recommendations to resolve them. Involves consultations with prescribers. Pharmacists utilize prospective and retrospective review methods in PACE: ○ Prospective MSRs address MRPs at prescribing-dispensing interface (prior to drug ingestion). ○ Retrospective MSRs address MRPs found in a pre-existing regimen (after drug ingestion). MSRs can be delivered telephonically or electronically (e.g., e-mail, instant message, or fax). In MSRs, pharmacists identify about 2 MRPs per PACE participant. About 80% of all MRPs in PACE involve DDIs (36%), ADRs (18%), high doses (14%), and unindicated medications (13%). MRPs are often resolved through deprescribing (25%), changing drugs (25%), or changing doses (20%). Prescribers accept nearly 80% of recommendations [11].

Abbreviations: ADE = adverse drug event; ADR = adverse drug reaction; CDSS = clinical decision support software; CPIC = Clinical Pharmacogenetics Implementation Consortium; DDI = drug interaction; ED = emergency department; MRP = medication-related problem; PACE = Programs of All-inclusive Care for the Elderly; PD = pharmacodynamic; PK = pharmacokinetic; TRHC = Tabula Rasa HealthCare.

**Table 2 healthcare-10-00551-t002:** Example contingency table.

	MRM	Control ^1^
≥1 ADE in 2018 but not in 2019	# participants	# participants
No year-over-year change in ADEs	# participants	# participants
No ADE in 2018 but ≥1 in 2019	# participants	*#* participants

Abbreviations: MRM = Medication risk mitigation; ^1^ proportions were risk-adjusted based on weights of HCC distribution.

**Table 3 healthcare-10-00551-t003:** Baseline demographics.

	MRM + Control	MRM	Control	*p*-Value ^1^
Participants, n (%)	2545 (100)	1582 (62.2)	963 (37.8)	N/A
Male, n (%)	834 (32.8)	537 (33.9)	297 (30.8)	0.11
Age, mean (95% CI)	77.0 (76.6, 77.3)	76.7 (76.2, 77.2)	77.4 (76.8, 78.1)	0.09
HCC score, mean (95% CI)	2.64 (2.59, 2.69)	2.68 (2.62, 2.74)	2.58 (2.50, 2.65)	0.042
Conditions, n (%)				
Hypertension (I10)	1460 (57.4)	973 (61.5)	487 (50.6)	<0.001
Diabetes, type II (E11)	1137 (44.7)	776 (49.1)	361 (37.5)	<0.001
Dyslipidemia (E78)	1046 (41.1)	659 (41.7)	387 (40.2)	0.23
Dementia (F03)	506 (19.9)	329 (20.8)	177 (18.4)	0.14
COPD (J44)	490 (19.3)	293 (18.5)	197 (20.5)	0.23
Major depressive disorder (F33)	436 (17.1)	300 (19.0)	136 (14.1)	0.002
Heart failure (I50)	144 (5.7)	81 (5.1)	63 (6.5)	0.13
Location of PACE, n (%)				
Northeast ^2^	859 (33.8)	177 (11.2)	682 (70.8)	<0.001
South ^3^	623 (24.5)	457 (28.9)	166 (17.2)	
Midwest ^4^	294 (11.6)	179 (11.3)	115 (11.9)	
West ^5^	769 (30.2)	769 (48.6)	0 (0.0)	

Abbreviations: COPD = Chronic obstructive pulmonary disease; HCC = Hierarchical condition category scores; MRM = Medication risk mitigation; ^1^ Nominal variables were compared with the chi-square test and continuous variables were compared with the independent *t*-test. ^2^ Massachusetts, New Jersey, and Pennsylvania. ^3^ Florida, North Carolina, and South Carolina. ^4^ Arkansas, Iowa, Michigan, and Oklahoma. ^5^ California and Colorado.

**Table 4 healthcare-10-00551-t004:** Year-over-year changes in medical expenditures adjusted by the actual capitated rate ^1^.

Group	2018, Mean (95% CI)	2019, Mean (95% CI)	Year-over-Year Change ^2^ (95% CI)	% Change (95% CI)	Weighted Mean Difference ^3^, Absolute	*p*-Value ^4^
Mean total medical expenditures per participant: combined facility and physician (US Dollars)						
MRM	USD 22,841 (USD 21,465, USD 24,218)	USD 27,228 (USD 25,664, USD 28,792)	USD 4386 (USD 3040, USD 5732)	19.2% (13.3%, 25.1%)	USD 5024	*t*: <0.001 W: <0.001
Control	USD 25,418 (USD 23,781, USD 27,055)	USD 34,829 (USD 32,873, USD 36,784)	USD 9410 (USD 7737, USD 11,084)	37.0% (30.4%, 43.6%)		
Mean physician expenditures per participant (US Dollars):						
MRM	USD 11,932 (USD 11,295, USD 12,570)	USD 13,800 (USD 13,064, USD 14,536)	USD 1868 (USD 1399, USD 2336)	15.7% (11.7%, 19.6%)	USD 1217	*t*: <0.001 W: <0.001
Control	USD 10,727 (USD 10,061, USD 11,394)	USD 13,811 (USD 13,003, USD 14,621)	USD 3085 (USD 2493, USD 3676)	28.8% (23.2%, 34.3%)		
Mean facility expenditures per participant (US Dollars)						
MRM	USD 10,909 (USD 9791, USD 12,027)	USD 13,428 (USD 12,165, USD 14,691)	USD 2519 (USD 1386, USD 3651)	23.1% (12.7%, 33.5%)	USD 3807	*t*: <0.001 W: <0.001
Control	USD 14,691 (USD 13,195, USD 16,187)	USD 21,017 (USD 19,088, USD 22,945)	USD 6326 (USD 4757, USD 7894)	41.3% (32.4%, 53.7%)		

Abbreviations: MRM = Medication risk mitigation; *t* = *p*-value from weighted *t*-test; W = *p*-value from Wilcoxon test. ^1^ The cost outcomes for 2018 and 2019 reported were adjusted by the actual capitated rate for each participant. Adjustments were applied to the control group’s 2018 and 2019 costs. This ensured that geographic differences between MRM and control did not bias outcomes. ^2^ 2019–2018 costs. ^3^ Year-over-year change for control–year-over-year change for MRM. ^4^ Comparison is between each group’s mean year-over-year change (weighted *t*-test) or median year-over-year change (Wilcoxon).

**Table 5 healthcare-10-00551-t005:** Year-over-year changes in clinical outcomes adjusted by hierarchical condition category scores ^1^.

Group	2018 (95% CI)	2019 (95% CI) ^1^	Year-over-Year Change, Absolute ^2^ (95% CI)	Year-over-Year Change, % (95% CI)	Weighted Difference ^3^, Absolute	*p*-Value ^4^
ADEs (fraction of participants with at least 1 ADE per year):						
MRM	0.068 (0.056, 0.081)	0.069 (0.056, 0.081)	0.001 (−0.015, 0.016)	0.9% (−21.4%, 23.2%)	0.023	χ^2^: 0.17
Control	0.055 (0.040, 0.071)	0.079 (0.060, 0.097)	0.023 (0.003, 0.043)	42.2% (5.7%, 78.7%)		
Falls (fraction of participants with at least 1 fall per year)						
MRM	0.11 (0.09, 0.12)	0.12 (0.10, 0.14)	0.013 (−0.007, 0.034)	12.4% (−6.8%, 31.9%)	0.016	χ^2^: 0.65
Control	0.11 (0.09, 0.13)	0.14 (0.12, 0.16)	0.029 (0.000, 0.058)	25.9% (0.1%, 51.7%)		
Emergency department visits (mean number of visits per participant per year)						
MRM	1.5 (1.4, 1.7)	1.6 (1.4, 1.7)	0.04 (−0.12, 0.19)	2.4% (−7.9%, 12.6%)	0.14	*t:* 0.20 W: 0.27
Control	1.9 (1.7, 2.2)	2.1 (1.8, 2.4)	0.17 (−0.06, 0.47)	9.1% (−6.2%, 24.4%)		
Hospital admissions (mean number of admissions per participant per year)						
MRM	0.32 (0.28, 0.35)	0.36 (0.32, 0.40)	0.04 (−0.01, 0.09)	12.7% (−1.6%, 27.1%)	0.025	*t:* 0.26 W: 0.17
Control	0.33 (0.28, 0.38)	0.40 (0.34, 0.46)	0.07 (0.001, 0.13)	19.6% (0.2%, 39.0%)		

Abbreviations: ADE = Adverse drug events; MRM = Medication risk mitigation; *t* = *p*-value from weighted *t*-test; W = *p*-value from Wilcoxon test; χ^2^ = *p*-value from chi-square test. ^1^ The outcomes reported were adjusted by hierarchical condition category scores. Adjustments were applied to the control group’s 2018 and 2019 outcomes. This ensured that differences related to multimorbidity between MRM and control did not bias outcomes. ^2^ 2019–2018. ^3^ Year-over-year change for control–year over year change for MRM. ^4^ Comparison is between each group’s change score.

## Data Availability

The data presented in this study are available on request from the corresponding author. The data are not publicly available due to privacy concerns.

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
