# Peer review of "Evaluating the Impact of Medication Risk Mitigation Services in Medically Complex Older Adults"

_healthcare, 2022, doi:10.3390/healthcare10030551_

Round 1

Reviewer 1 Report

Dear authors,

Thank you for the work. These are my comments:

Brief summary

This manuscript was designed with the aim to determine the impact of  Program of All-Inclusive Care for the Elderly  PACE on the total healthcare cost and other outcomes among the elderly subjects as compared to the control population (without PACE). The study found out that the implemented service (MRM in PACE) was significantly reduced in the healthcare cost, but not for the other types of outcomes eg adverse drug events, hospital admission and emergency department visit.

Introduction

It would be good if more details on the PACE and MRM were explained in the introduction, so the readers will have a clear idea on what constitute PACE and MRM and the background of the service.

Method

  • The whole description of PACE and MRM are more suitable to be in the introduction section than the method. The authors may focus on the method of intervention or study population specific to the study, rather than the general statement on the population included in the PACE with the objectives etc.
  • It would be good to include the ethical approval ID in the manuscript.
  • Sampling method is lacking in the method section. Please explain accordingly.
  • Sample size calculation is lacking in the manuscript. Please explain accordingly.
  • More information on how the cost was calculated would be good, including the formula used and examples of facility and physician expenditure.

Table

Table 3: the results for t-test is refereed to which comparison? An explanation or label in the footnote would be helpful to increase the understanding

Reviewer 2 Report

Thank you for submitting this article. This is a very interesting manuscript, and I commend you for the robust analysis. I think the background on the PACE program would best fit in the introduction, as readers unfamiliar with this program may need more information before diving into the methods. 

Did you examine a change in medication costs? It would be interesting in a future manuscript to evaluate the type of interventions pharmacists performed with this patient group; what were the most common interventions?

In the methods section, I would like to know more about how pharmacists are altered to perform MRMs. Are all patients eligible? Does the program identify select patients? Are reports generated? I think there also needs to be more information on how MRMs are performed, as this process may be new for some readers. 

I like the focus on ADEs, but based on the confounding variables, it may be better to focus on why MRMs are important in the introduction. ADEs are just one piece of the picture when we discuss medication use in this population. 
